

# A 500-year multi-proxy drought reconstruction for the Czech Lands

Petr Dobrovolný[1,2], Rudolf Brázdil[1,2], Miroslav Trnka[2,3], Michal Rybníček[2,4], Tomáš Kolář[2,4], Martin Možný[5], Tomáš Kyncl[6], Ulf Büntgen[1,2,7,8]

[1]Department of Geography, Masaryk University, Brno, Czech Republic
[2]Global Change Research Institute, Czech Academy of Sciences, Brno, Czech Republic
[3]Department of Agrosystems and Bioclimatology, Mendel University, Brno, Czech Republic
[4]Department of Wood Science, Mendel University, Brno, Czech Republic
[5]Doksany Observatory, Czech Hydrometeorological Institute, Doksany, Czech Republic,
[6]Moravian Dendro-Labor, Brno, Czech Republic
[7]Department of Geography, University of Cambridge, United Kingdom
[8]Swiss Federal Research Institute for Forest, Snow and Landscape WSL, Birmensdorf, Switzerland

*Correspondence to*: Petr Dobrovolný (dobro@sci.muni.cz)

**Abstract.** Any proxy archive related to climate has inherent advantages and disadvantages. What have become known as the
"multi-proxy approaches" therefore constitute the cutting edge of paleoclimatology, as they are capable of providing more complete pictures of past climatic changes. This contribution combines tree-ring width chronologies, grape harvest dates and documentary-based precipitation indices from the territory of the Czech Republic to reconstruct inter-annual to multi-decadal drought variability, as expressed by Z-index, back to AD 1501. Using Principle Component Analysis and simple linear regression, the multi-proxy record explains 70% of April–July drought variability between 1805 and 1854. It is demonstrated
that the relatively short calibration period of early instrumental measurements, from 1805 to 1854, does not influence the quality of this reconstruction, and that the regression approach does not underestimate drought extremes. While reflecting a considerable amount of inter-annual spring-summer drought variability, the new Z-index reconstruction does not capture any long-term trends beyond the multi-decadal domain. The driest (1616) and the wettest (1713) years match previously published hydroclimate reconstructions from the same region, whereas the driest and wettest 30-year periods occurred in
1998–2017 and 1890–1919 respectively. Thus central Europe has recently experienced the most severe decadal-scale late spring–early summer drought of the past 500 years. The new multi-proxy drought reconstruction demonstrated progress beyond previous single-proxy attempts at establishing the strength of hydroclimate signal.

## 1 Introduction

Better understanding of past and present hydroclimate variability is of great importance in the light of the known negative
impacts of drought on nature and society (Ciais et al., 2005; Trenberth et al., 2014; Wilhite and Pulwarty, 2018). Problems related to the possible intensification of hydroclimatic extremes and the rising risk of drought occurrence are rendered still more pressing by the degrees of uncertainty associated with model projections of future hydroclimate compared to future



temperature conditions (Dai, 2013; IPCC, 2013; Greve et al., 2018). It is anticipated that events such as the droughts of 2003 (Brázdil and Trnka, 2015) and 2015 (Hoy et al., 2017; Laaha et al., 2017) will became twice as frequent by the end of the 21st century (Samaniego et al., 2018). However, the exceptionality of such events is largely considered within the scale of the relatively short instrumental period. Both model uncertainties and unclear exceptionality of extremes in the longer-term

context may be partly corrected with hydroclimate proxy reconstructions.

Past hydroclimate variability has become subject to intensive study at both the global (Ljunquist et al., 2016; Steiger et al., 2018) and continental levels (PAGES Hydro2k Consortium, 2017; Linderholm et al., 2018; Markonis et al., 2018). High spatio-temporal hydroclimate variability suggests the need for analyses at more regional and local scales (Oberhuber and Kofler, 2002; Čufar et al., 2008; Tejedor et al., 2016; Karanitsch-Ackerl et al., 2017; Seftigen et al., 2017).

Long series of several precipitation-sensitive proxies have been collected for the Czech Lands, recently the Czech Republic (CR). They cover the last 500 years, at the least, and are derived either from natural archives such as tree rings (Brázdil et al., 2002; Büntgen et al., 2011; Dobrovolný et al., 2018) or from man-made archives containing documentary evidence (Dobrovolný et al., 2015; Možný et al., 2016b). Moreover, Brázdil et al. (2013b) have characterized drought occurrence in the Czech Lands for the last millennium by using documentary evidence and instrumental data. Documentary evidence and

instrumental measurements have been used to calculate several drought indices from AD 1501 onwards (Brázdil et al., 2016). Some of the above proxies from the CR have been used for quantitative reconstruction of drought/precipitation variability and have already demonstrated good reconstruction skills. However, individual proxies entail their own strong and weak features and may differ in many respects, such as sampling region, sensitivity to a range of variables, and seasonality. As a result they demonstrate some similarities but also distinct differences in reconstruction of past hydroclimate

variability. Lower coherence among proxies results in higher uncertainties for some periods.

To overcome some of the problems of the single-proxy approach, to identify possible seasonal biases and finally to provide more rigorous estimates of past and present hydroclimate variability, this contribution attempts to synthesize available precipitation-sensitive proxies from the territory of the CR and to utilize them in the first multi-proxy reconstruction of droughts in the area. First, the main features of the study area and data sources are introduced. Then methods and a

presentation of the resulting reconstruction are provided, together with uncertainty estimates. Finally, the main features of the new reconstruction are validated and discussed with respect to existing reconstructions for central Europe.

## 2 Study area and data

The territory of the CR is located in central Europe between 48°33'–51°03' N and 12°05'–18°51' E, with elevations ranging from *c*. 115 to 1600 m asl (Fig. 1). The climate is characterised by transition between maritime and continental influences,

with four annual seasons clearly expressed. Mean annual temperature varies between *c*. 5.5°C and 9°C. July is the warmest month (12°–20°C) while January is the coldest (-6°–0°C) depending on altitude. Precipitation distribution over the area features maximum totals in summer (JJA; June or July) and minimum in winter (DJF; January or February). The driest areas,




north-western Bohemia and southern Moravia, have a mean annual precipitation of 400–450 mm, whereas this value rises to 1600 mm in mountain regions in the borderlands (Tolasz et al., 2007).

Drought as target data is represented by Z-index for the purposes of this study. The Z-index (or Palmer moisture anomaly index) is an intermediate product in the calculation of the Palmer Drought Severity Index (PDSI; Palmer, 1965). Because

moisture conditions in previous months are not considered in its calculation, the Z-index reacts immediately to changes in soil moisture, therefore characterising short-term drought particularly well (Karl, 1986; Brázdil et al., 2015). A self-calibrated Z-index (Wells et al., 2004) was deployed in this study. This adjusts the empirical constants used in the original Palmer calculation on the basis of meteorological data from the study area. Mean Czech areal temperatures and precipitation totals for the 1805–2012 period, compiled by Brázdil et al. (2012) and extended up to 2018 were used for Z-index

calculation.

Altogether four different moisture-sensitive proxies, representing both natural and anthropogenic archives from the Czech territory, were employed in this study:

(i) The silver fir (*Abies alba* Mill.) ring width chronology (FIR) from South Moravia (the south-eastern part of the CR) consists of 117 living and 165 historical samples and goes back to AD 1325. The measurements replicate well after AD

1500. Büntgen et al. (2011) analyzed these data to evaluate the capacity of recent tree growth to track hydroclimate variability represented by Z-index for May–June. They revealed robust signal strength through both the living and historical samples. Standardized tree-ring widths (TRWs) were calibrated to instrumental measurements from the Brno meteorological station over the 1805–1932 period. The highest correlation between fir TRWs and Z-index (0.54) was obtained for the time before 1905, while it was significantly lower later.

(ii) Oak (*Quercus* spp.) ring-width chronology (OAK), consisting of measurements from more than 3,500 living, historical and sub-fossil trees covering the period from AD 352 to the present. These were compiled from 387 locations within the CR (Kolář et al., 2012; Prokop et al., 2017). Majority of samples of the two oak species (the English oak *Quercus robur* L. and the sessile oak *Quercus petraea* (Matt.) Liebl.) were collected in lowlands of central Bohemia (western part of the CR) and South Moravia (the south-eastern part of the CR). Dobrovolný et al. (2018) calibrated standardized TRW indices to spatially-

relevant grids from the CRU precipitation database. The oak TRW chronology explained 34% of May–July precipitation variability.

(iii) Grape harvest dates (GHD) were collected for the Bohemian wine-growing region (mainly north-west of Prague). GHDs were compiled from various documentary sources covering the 1499–1845 period (Možný et al., 2016b). For 1845 onwards, phenological observations of the Czech Hydrometeorological Institute were used. The GHD series was calibrated against the

mean Standardized Precipitation Evapotranspiration Index (SPEI) and then used for April–August SPEI reconstruction from 1499 to 2012. The SPEI reconstruction explained 75% of the SPEI from the instrumental period.

(iv) Monthly precipitation indices (PRE) for the spring–summer months were derived from documentary evidence in the territory of the CR for the 1501–1854 period. Information from narratives, non-instrumental weather diaries and similar sources were interpreted to indices ranging from −3 (extremely dry) to +3 (extremely wet) for each month. Missing values





for the CR were filled in with similar indices derived for Germany (Glaser, 2008) and Switzerland (Pfister, 1999). The most complete CR indices were available for summer. Subsequently, Dobrovolný et al. (2015) calibrated seasonal and annual index series to mean Czech precipitation series for the 1805–2010 instrumental period (Brázdil et al., 2012). Using simple linear regression, the index series shared 35% of JJA and 33% of MAM variance with the instrumental target data.

**3 Methods**

In the first step, the four proxy series were adjusted to the common period of 1501–1854 and standardized to z-scores. Because the instrumental target data cover the 1805–2018 period, correlation and regression analysis were used over the overlapping 1805–1854 period to test the degree of shared variability among different proxies. To investigate whether the quality of proxies significantly differs over time, correlation analysis was also performed for running 30-year periods over the entire lengths of the proxy series.

The importance of the individual proxy series as predictors for hydroclimate reconstruction was analyzed by means of step-wise regression using the forward selection model. This method fits various regression models to a set of possible predictors. In each step, individual predictors are added to the model or subtracted from the it. The relative quality of each model with a concrete set of explanatory variables is evaluated according to pre-selected criteria (Wilks, 2006). Here, the Akaike information criterion (AIC) was utilised as a relative measure considering two model parameters: the logarithm of the sum of residual squares and number of fitted variables. AIC tends to penalize models with higher numbers of variables and gives preference to relatively simple models. The most appropriate model is that with the minimum AIC value. Moreover, this approach is able to identify those variables that do not contribute significantly to overall explained variance.

The common signal strength of important predictors was evaluated by application of the mean correlation coefficient (R-bar) with Expressed Population Signal (EPS). EPS measures the statistical quality of a set of predictors with respect to hypothetical perfect (noise-free) and infinitely-replicated proxies (Briffa and Jones, 1990). EPS is used in this study for assessing the representation of the population signal. However, the generally accepted threshold (0.85) is chosen arbitrarily (Wigley et al., 1984; Buras, 2017).

The most important proxies from the step-wise regression were transformed using Principal Component Analysis (PCA) and the principal component (PC) scores used as a synthesis of individual proxies. Correlation (response) analysis then identified significant PCs and optimal season (combination of months) for calibration. PC scores were calibrated to Z-index values for the entire overlapping period (1805–1854) using the least-square regression.

 The strength of the drought signal in the calibration period was assessed by the square of Pearson's correlation coefficient ($R^2$), Root Mean Square Error (RMSE), and Durbin–Watson statistics (DW). Their interpretation can be found elsewhere (Wilson et al., 2013).

A characteristic feature of the linear regression method is an artificial reduction of variance of predicted values. This results in underestimation of the variability of the reconstruction (Esper et al., 2005). Therefore a second approach to calibration





was taken, a simple method based on mean and variance adjustment (scaling) of proxy data. The mean of the target data was added to the PC scores to ensure the same mean for both proxy and target. Consecutively-centred proxy data were multiplied by the target standard deviation to guarantee the same variance of proxy and target. Thus variance-scaled reconstruction captured the target variance better than the regression. However, this method produces a larger Mean Square Error (MSE)

which, in contrast, is minimized in the regression method. Both methods were compared with the Equivalent variance explained ($R_{vs}^2$), a metric suggested by McCarroll et al. (2015) that quantifies the magnitude of MSE increase proportional to the loss of climate signal when using variance scaling instead of regression-based calibration.

Calibration and verification were performed twice for the early (1805–1829) and late (1830–1854) sub-periods (for more details see e.g. Wilson et al., 2013; Dobrovolný et al., 2015). Values of $R^2$, reduction of error (RE), and the coefficient of

efficiency (CE) constituted the verification statistics for the regression method of calibration. RE and CE measure a reconstruction skill of proxy data. Reconstructed values are compared to a hypothetical reconstruction that is represented simply by the mean of the target data in calibration (RE) or verification (CE) periods. Positive RE (CE) values are generally acceptable (see e.g. Cook et al., 1994; Wilson et al., 2006). Corresponding measures for $R_{vs}^2$, $RE_{vs}$, and $CE_{vs}$ were calculated for the variance-scaling method.

Final reconstruction was completed with uncertainty estimates (error bars). Two different sources of error were considered for their construction. The first was the regression error defined as two (1.96) standard errors of estimate from the corresponding regression model, approximating a 95% uncertainty estimate. The problem is that the regression error quantifies only a part of the overall uncertainty as it is based on data from the relatively short overlapping period. The second source of uncertainty was estimated from the mean inter-series correlations between all pairs of proxies (R-bar). The R-bar

expresses the quality of proxies included in the analysis. The use of R-bar for the adjustment of uncertainty estimate arises out of the idea that higher inter-series correlation (high R-bar) demonstrates higher quality of data and lower uncertainty in reconstruction, and *vice-versa* (Dobrovolný et al., 2010; Leijonhufvud et al., 2010).

## 4 Results

### 4.1 Multi-proxy synthesis and proxy climate response analysis

As follows from Fig. 2a, four proxy series show large decadal-scale variability and they do not contain any long-term tendencies. This is typical of OAK and PRE series in particular. In the GHD series, below-mean values prevailed during the 16th century and especially after the 1970s. For the FIR series, a well-expressed period of wider variability marks the period after the 1950s. Conversely, two periods of distinctly lower variability, centred approximately on the 1650s and the 1750s, may be identified in the GHD series. Relatively stable variability is typical of the OAK and PRE series.

It is notable that the strength of the common signal expressed by mean correlation (R-bar, Fig. 2b) shows a decreasing trend overall. This means that the common variance included in the four proxies (the signal) is markedly stronger in the 16th–18th centuries and distinctly weaker in the 19th–20th centuries. A shorter period of lost common coherence also occurred around




the 1750s. EPS values confirm the same characteristics of the proxies in terms of their quality. While they are above or close to the 0.85 threshold before the 1820s, they decrease thereafter. This is related to lower mutual correlations between proxies and also to the fact that, from the 1850s onwards, documentary indices with relatively strong hydroclimate signal are absent. Fig. 2b also summarizes overall correlations between individual proxies for the common period of 1501–1854. All these

correlations are statistically significant. The highest correlation appears between oak TRWs and precipitation indices, while the lowest occurs between fir TRWs and GHD. The GHD series shows lower correlations with the two TRW series.

Further, progressive selection of the original proxies OAK, FIR, GHD, and PRE and their various combinations against Z-index values in the multiple regression model was undertaken. According to AIC, the inclusion of the four proxy series provides the best model, with a highly significant proportion of explained common variability in terms of adjusted $R^2$

10   (67.8%).

The results of the multiple regression model, the sensitivity of the four proxy series to various measures of hydroclimate demonstrated in previous studies (see Sect. 2), and a clear correlation structure among them (Fig. 2b) indicated that they could be transformed by means of PCA. This approach facilitated the removal of redundant information from the original variables and description of hydroclimate variability with the PCs as new (synthetic or latent) "proxies". PC scores explained

55% (PC1), 76% (PC1+PC2), and 90% (PC1+PC2+PC3) of the cumulative variability of the four original proxy series. Subsequently, the PCs were tested against Z-index values in the 1805–1854 overlapping period. The results summarized in Fig. 3 demonstrate that PC1 clearly dominates in all month combinations and April–July (AMJJ) appears as the optimal season for Z-index reconstruction, with Pearson's correlation coefficient reaching 0.83, indicating 69.7% of explained variance between PC1 and Z-index target data.

**4.2 Calibration and verification results**

The overview of statistics that appears in Table 1 demonstrates quite a high reconstruction skill for PC1 scores, not only over the entire overlapping period (1805–1854), which was used for the final calibration, but also for the shorter sub-periods used for validation. The linear regression model explained almost 70% of common variance. The DW statistics provide acceptable quality of the regression model. Calibration with the variance scaling approach provides very similar results compared with

the regression method, as is evident not only in Table 1 but also from direct comparison of measured and reconstructed Z-index values (Fig. 4). Comparison of $R^2$ and $R_{vs}^2$ demonstrates negligible loss of signal for the early, late, and full calibration periods when the variance-scaled method of calibration is used. The slight difference between $R^2$ and $R_{vs}^2$ also indicates that bias towards the mean and reduced variance due to regression in this reconstruction are quite small.

All verification statistics (RE, CE, $RE_{vs}$, $CE_{vs}$) are similar and highly positive. Reconstructed Z-index values approximate

measured data well in most of the years and also clearly reproduce their inter-annual variability independent of method and the calibration sub-periods involved (Fig. 4). This demonstrates that the hydroclimate signal included in the PC1 scores transformed from the original four proxy series is strong and stable over time, at least during the first part of the 19th century.



### 4.3 Z-index reconstruction

The final April–July Z-index reconstruction for the 1501–1854 period was spliced with measured data from 1805 to 2018, thus providing a chronology of more than 500-years for the territory of the CR. The chronology shows considerable high-frequency variability; no clear long-term trend is evident (Fig. 5).

The 95% confidence level of the reconstruction in Fig. 5 was defined as explained in Section 3. As Fig. 2 indicates, the quality of proxies expressed with R-bar for the period before AD 1800 was quite stable and surprisingly higher than afterwards, in the calibration period and during the 20th century. This means that there is no reason to adjust error estimates in the light of the lower quality of proxy data during the 16th–18th centuries. The only period with lower coherence among proxies used is centred around the 1740s (Fig. 2). No specific feature in the individual chronologies used, such as for data

quality, number of replications, etc. was found. Interestingly, a similar dip in correlations emerged in a central European temperature reconstruction from documentary data (see Fig. 13 in Dobrovolný et al., 2010).

### 5 Discussion

It is evident from Fig. 3 that significant correlations were found not only for April–July, but also for other combinations of spring–summer months. This may be a reflection of the slightly different seasonality of the original proxy series, as occurs,

for example, in May–June for fir TRWs (Büntgen et al., 2011) and April–August for GHD (Možný et al., 2016b). This feature to some extent demonstrates the robustness of the new Z-index reconstruction, since this allows compilation of drought reconstructions for slightly different combinations of months (seasons) of comparable quality measured via the total of explained common variance (Fig. 3). Moreover, this feature could allow a direct comparison with similar existing hydroclimate reconstructions without substantial loss of coherence due to varying seasonality.

### 5.1. Czech Z-index reconstruction in the European context

The driest year (1616) had already been identified in numerous studies analyzing documentary evidence (Brázdil et al., 2013b; Dobrovolný et al., 2015) and tree-ring chronologies (Dobrovolný et al., 2018) covering CR territory. That the summer months of 1616 were extremely hot and dry has also been disclosed from documentary evidence from Germany (Glaser, 2008) and Switzerland (Pfister, 1999). Considerable variability of Z-index can be demonstrated for the 1710s,

during which the extremely wet year of 1713 was soon followed by two very dry years in 1718 and 1719 (Brázdil and Trnka, 2015). The year of 1713 was also identified as exceptionally wet by Büntgen et al. (2011) in May–June Z-index reconstruction for South Moravia (south-eastern CR). Extremely dry 10-year and 30-year periods exhibit higher variance compared to wet ones, and a similar asymmetry also emerges in absolute extremes. The driest year (1616) deviates from the mean much farther than the wettest (1713).

The decade 1532–1541 was the driest, corresponding with the very warm and dry 1530s previously noted (Brázdil et al., 2013a), culminating in Europe in the extremely warm and dry year of 1540 (Wetter and Pfister, 2013; Wetter et al., 2014).



The wettest decade occurred in 1763–1772. The driest 30-year period fell in the most recent decades (1988–2017) and clearly corresponds to rising temperatures, while at the same time precipitation totals over the territory of CR do not show any statistically significant trends (e.g. Brázdil et al., 2012). On the other hand, a clearly-expressed drought spell in the 1780s–1810s corresponds with very low mean precipitation totals, especially during MAM and partly also in JJA over the

territory of the CR (see Fig. 7 in Dobrovolný et al., 2015).

Aside from the recent driest period (1998–2017), smoothed data identify at least three relatively long periods of below-mean values centring approximately on the mid-16th century, the beginning of the 17th century, and the period covering the late 18th–early 19th centuries. On the other hand, with the single exception of the wettest 1890–1919 spell, correspondingly longer periods of above-mean values are not evident in the smoothed chronology. This demonstrates a tendency to more

persistent drier periods compared to wet spells. The same feature appeared in the analysis of scPDSI on a European scale by Marconis et al. (2018). However, according to these authors, wet conditions prevailed during the last 90 years and a recent increase in wetness is also evident in central Europe. This is in clear contradiction to the decreasing trend in reconstructed Z-index series in this study, as well to other drought-related studies in the CR based on drought indices (Brázdil et al., 2009, 2016; Brázdil and Trnka, 2015).

Data-independent reconstructions of short-term droughts for central Europe are scarce. Similar reconstructions refer either to long-term drought (Cook et al., 2015) or to precipitation regime (Wilson et al., 2005; Pauling et al., 2006). Other available hydroclimate reconstructions, presented e.g. in Fig. 9 of Dobrovolný et al. (2018), share some part of the proxies used herein or they represent climatologically distinct regions. The new Z-index series was compared with the regionally-relevant part of the Old World Drought Atlas (Cook et al., 2015). Differences between the Z-index and scPDSI were partly suppressed by

low-pass filtering of the original series (Fig. 6a). Both series show good agreement for a substantial part of the 16th century and the second parts of the 18th and 19th centuries. However, they are out of phase in recent decades, when the Z-index demonstrates a sharp drop, which is not evident in scPDSI. Running correlations (Fig. 6b) confirm our findings from the analysis in Sect. 4.1. While there are highly significant correlations from the 16th century up to the 1850s, coherence is much lower thereafter (see also Fig. 8 and discussion in Sect. 5.2).

**5.2 Reconstruction uncertainties**

It follows from the temporal coverage of the four basic proxies considered that inclusion of the precipitation indices based on documentary evidence (PRE), which are available only before 1855, may be limiting for two reasons: (i) this significantly reduces the length of the overlapping period to only 50 years); (ii) this relatively short period covers the early instrumental period of the first part of the 19th century for which measured data are limited to only a few stations.

To minimize the risk that the calibration/verification results herein were significantly influenced by a relatively short overlapping period, a random calibration approach was adopted. A random selection of 25 years (without replacement) from the full 1805–1854 period of the overlap was made and calibrated to the corresponding Z-index values; a complemented 25





years were used for validation. The whole process was repeated 500 times for linear regression and for variance scaling methods. Verification statistics were calculated for each trial. The results are summarized in Fig. 7.

The distribution of the verification statistics shows strong positive values for regression and variance-scaling methods of calibration (Fig. 7). The variance explained ($R^2$) between proxy and target data fluctuates on average from 64% to 72% for

the various 500 trials of random calibration. Further, RE, CE, $RE_{vs}$ and $CE_{vs}$ values reach, on average, 0.65–0.70 with slightly lower values for variance-scaling. The mean difference between $R^2$ for regression and variance-scaling does not exceed 5%. These results demonstrate two important features: (i) the set of four proxies integrated in PC1 has very strong reconstruction skill for the 1805–1854 calibration period; (ii) the regression-based calibration reduces the real variability of reconstructed Z-index values only slightly (Bürger et al., 2006; McCarroll et al., 2015).

The quality of target data from the first part of the 19th century was evaluated with mean CR temperature and precipitation series (Brázdil et al., 2012) as the primary data source for the calculation of Z-index series in the instrumental period. These mean series were compiled from long-term monthly temperature and precipitation series from selected meteorological stations. The number of stations covering the first part of the 19th century before 1854 was lower (5 temperature series and 4 precipitation series) compared to the 20th century (10 temperature series and 12 precipitation series). All series from the

individual stations were homogenized using the PRODIGE method (Brázdil et al., 2012). The algorithm used for compilation of the mean CR series considered the fact that the variance of a mean series computed from a lower number of input series is reduced compared to variance of a mean compiled from larger set of input series. Thus the quality of the Z-index measured series (target data) before 1854 is comparable to the quality of the rest of the series after that time.

The problem of a relatively short calibration period could be solved only if PRE data were to be excluded from the analysis;

the reconstruction skill of the proxies was therefore tested without documentary indices. First, the results for 1805–1854 period with and without the PRE series were compared and it was found that the reconstruction skill measured by $R^2$ achieved only 59% when PRE data were excluded. This is a significantly lower value than the explained common variance between PC1 and Z-index when all four proxies – as high as 70% – were used (see Table 1).

In the next step, results of the reconstruction based only on OAK, FIR and GHD were evaluated, but for different calibration

periods, since the far longer 1805–2007 overlapping period was available, when PRE data were excluded. However, testing different 25- and 50-year-long calibrations obtained neither better reconstruction skill measured by $R^2$ nor better verification statistics than those summarized in Table 1. For instance, OAK, FIR and GHD proxies transformed via PCA and calibrated to Z-index in the most recent 1951–2000 period gave $R^2$ of only 25% and the verification statistics failed to provide positive values of CE.

Experiments with OAK, FIR and GHD proxies and with longer and the more recent calibration/verification periods demonstrated the high added value of documentary data for the final reconstruction. Furthermore, worse calibration results compared to those from the first part of the 19th century also highlighted the problem of lost coherence in the proxy–climate response analysis. As is evident from Fig. 2, whereas the individual proxy series used are largely in phase until approximately the mid-19th century, they exhibit quite opposite tendencies in some cases after that time. The most evident



differences concern FIR and GHD series in the most recent decades. Fir growth in central Europe during the 20th century has been influenced by numerous environmental changes related, for example, to air pollution and to acid-rain occurrence leading to "fir die-off" (Vrška et al., 2009). Successive air-quality improvements (Filipiak and Ufnalski, 2004), together with a warmer, but not drier, climate (Büntgen et al., 2014; Bosela et al., 2016) may be responsible for regeneration and a recent

positive growth reaction in fir trees. On the other hand, a strong decline in grape harvest dates (i.e. earlier vintages) is directly related to recent rising temperatures (Možný et al., 2016a). These examples demonstrate that the sensitivity of FIR and GHD series to other meteorological elements or environmental processes may obscure their drought signals when these processes start to dominate.

The problem of lost coherence in the proxy–climate response in the 20th century has already been noted in several

hydroclimate studies. It has been discussed in Büntgen et al. (2011) for fir TRW from South Moravia; Wilson et al. (2013) for oak TRW from southern-central England; Prokop et al. (2016) for oak TRW from Slovakia and Dobrovolný et al. (2018) for oak TRW from western CR. Trnka et al. (2015) suggest that the weaker response of TRW to hydroclimate may be related to the decreasing late MAM–early JJA soil moisture content in the CR. This may partly explain very low correlations between Z-index and the two TRW series used in this study from the 1980s. However, the problem appears to be more

complex, as outlined in Fig. 8, in which temporal changes in measured Z-index and the four proxy series are presented. There are two distinct drops in mean correlation (R-bar), in the 1860s–1880s and in the 1960s–1980s. Whereas the lost coherence in the latter period may be related to the above decline in JJA soil moisture, no particular reason for the 1860s–1880s is clear. Moreover, the GHD series exhibits quite different behaviour from the mean pattern outlined. The evident decline in its correlation with Z-index is shifted to the last decades of the 19th century and then it increases to significant

values. This is related to the fact that GHDs are also highly correlated to temperatures (see Možný et al., 2016a).

As the previous results demonstrated the importance of the documentary-based precipitation indices, the value of the similar temperature indices were tested as an additional explanatory variable in the drought reconstruction. Temperature indices were derived from documentary evidence for the territory of the CR, Switzerland and Germany, which characterise the temperature variability of central Europe well over the past 500 years (Dobrovolný et al., 2010). Because drought is usually a

combination of above-mean temperatures and below-mean precipitation, temperature indices may contain a distinct hydroclimate signal. Moreover, temperatures are used, as well as precipitation, for calculation of various drought indices such PDSI (Palmer, 1965) and SPEI (Vicente-Serrano et al., 2010). Despite these facts, the inclusion of temperature indices into the step-wise regression analysis did not improve the regression model significantly (not shown).

### 5.3 Ability to reproduce extremes

Finally, the Z-index reconstruction was tested whether it is able to preserve the driest and the wettest years using an approach suggested by McCarroll et al. (2015). The overlapping period 1805–1854 was examined and 10% of the years (i.e. five) from both tails of the Z-index distribution were defined as extremes. The instrumental and reconstructed Z-index series were sorted in ascending order and numbers of corresponding „reconstructed" and „instrumental" extremes were compared



(Fig. 9). While the regression-based reconstruction captured three extremely dry years and one extremely wet year, the variance-scaling reconstruction reproduced three dry and two wet extremes. This means that: (i) both reconstruction methods perform quite similarly in this respect; (ii) the figures for dry extremes show no significant difference (p <0.05) between the number of corresponding extremes found in the reconstructed and target Z-indices according to the extreme-value capture test after McCarroll et al. (2015).

**5 Conclusions**

Careful selection of available moisture-sensitive proxies and numerous calibration/verification trials resulted in a new chronology of short-term drought over the CR during the last 500 years. It consists of a synthesis of four different proxies and its high reconstruction skill demonstrates the clear advantage of a multi-proxy approach. The new chronology of Z-index spliced from reconstructed (1501–1854) and instrumental (1805–2018) parts shows that central Europe experienced the most severe 30-year late spring–early summer period of drought for the last 500 years during the most recent decades (1988–2017).

This analysis unequivocally demonstrates the value of documentary data but also constitutes an alert to the problem of lost coherence in the relationship between drought-sensitive proxies and climate in recent decades. This feature has already emerged in a number of studies and appears to be a rather complex issue with no single cause; multiple causes must be sought. This new Czech drought reconstruction, with quite high reconstruction skill derived from four proxies in the first part of the 19th century, indicates that the problem of lost coherence in the 20th century may be related to recent climate change and to the reactions of ecosystems to rapidly changing temperatures.

**Data availability.** The series of proxies used in the paper for reconstruction are available from the corresponding authors/publications.

**Competing interests.** The authors declare that they have no conflict of interest.

**Acknowledgements**. M.T., M.M. and U.B acknowledge the financial support of the SustES – Adaptation strategies for sustainable ecosystem services and food security under adverse environmental conditions project no. CZ.02.1.01/0.0/0.0/16_019/0000797. R.B. and P.D were supported by the Czech Science Foundation project no. GA17-10026S; M.R. and T.K. were supported by the GA18-11004S project. Tony Long (Svinošice) helped work up the English.

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



**Table 1. Summary of calibration and verification statistics used for AMJJ Z-index reconstruction for the territory of the CR using linear regression and variance scaling (VS); see text in Section 3 for explanation of individual statistics**

| Sub-period | $R^2$(%) | RMSE | DW | RE | CE | $R_{vs}^2$(%) | $RE_{vs}$ | $CE_{vs}$ |
|---|---|---|---|---|---|---|---|---|
| Early calibration (1805–1829)/ late verification (1830–1854) | 69 | 0.56 | 1.6 | 0.69 | 0.69 | 66 | 0.64 | 0.63 |
| Early verification (1805–1829)/ late calibration (1830–1854) | 71 | 0.55 | 2.0 | 0.68 | 0.67 | 69 | 0.67 | 0.67 |
| Full calibration (1805–1954) | 70 | 0.55 | 1.9 | – | – | 67 | – | – |



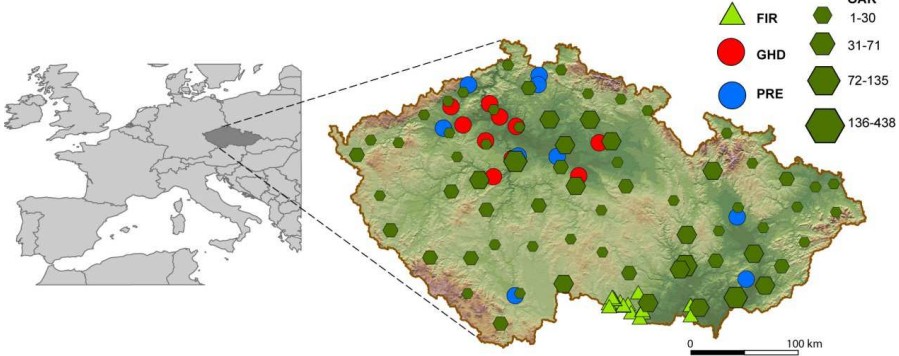

**Figure 1. Study area, showing the distribution of sampling sites of the fir TRW measurements (FIR), number of oak TRW samples in the CR districts (OAK) and localities that most significantly contributed to compilation of series of grape harvest dates (GHD)**
5   **and precipitation indices (PRE)**



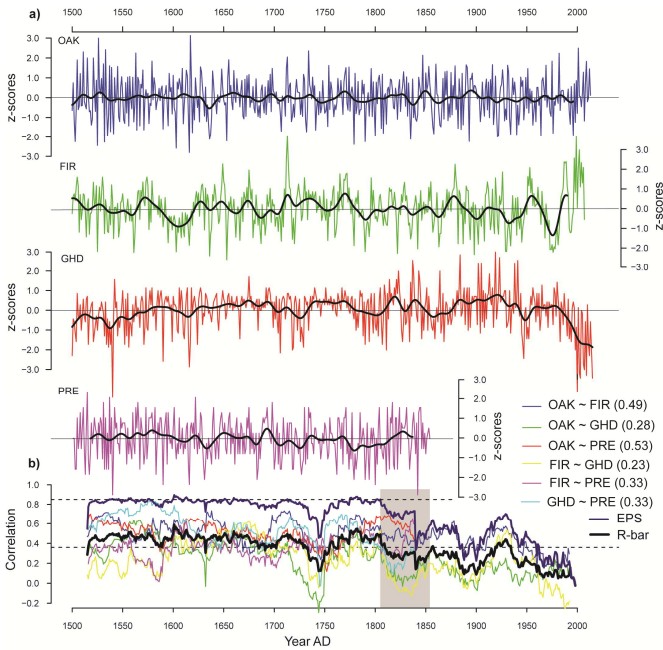

**Figure 2. (a) Variability of four individual proxies (expressed as z-scores) smoothed by low-pass filter; (b) 30-year running correlations between four individual proxies completed with mean running correlation (R-bar) and EPS. Dashed lines denote significant correlations and an acceptable EPS value (0.85); the grey area marks the overlapping period with target data (1805–1854). Numbers in brackets are Pearson correlation coefficients between individual proxies over the 1501–1854 period.**





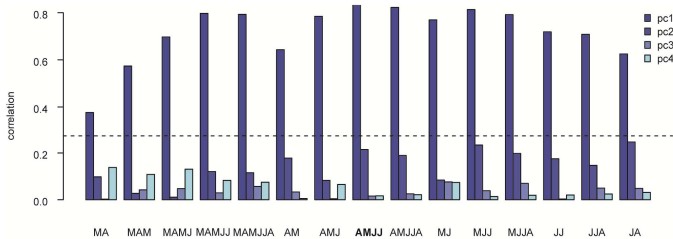

**Figure 3. Pearson's correlation coefficients between PC1–PC4 and Z-index for combinations of months in the 1805–1854 period; all correlations were transformed to positive values for clearer interpretation; the horizontal line identifies values above which correlations are statistically significant (n = 50, p <0.05)**




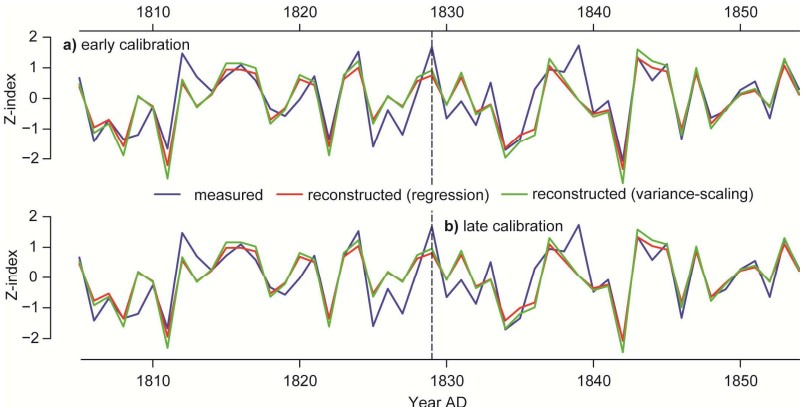

**Figure 4. Measured (blue) and reconstructed (red – linear regression, green – variance-scaling) AMJJ Z-index anomalies (1961–1990 reference period) in the CR for (a) early calibration (1805–1829) / late verification (1830–1854) and (b) late calibration (1830–1854) / early verification (1805–1829)**





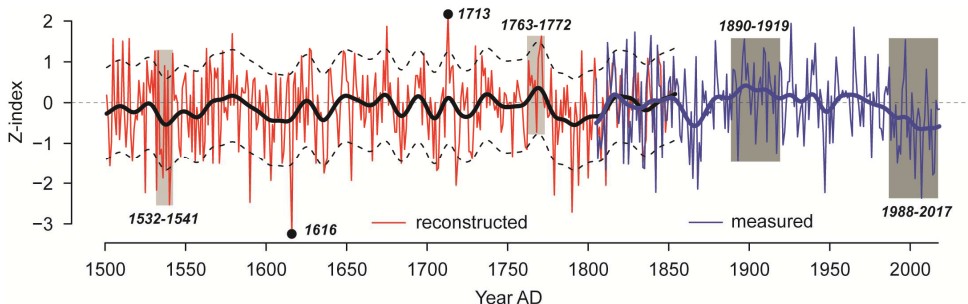

**Figure 5.** Fluctuations in reconstructed (1501–1854) and measured (1805–2018) April–July Z-index series for the territory of the CR smoothed by 30-year Gaussian filter (bold) and error bands (dashed) approximating 95% confidence intervals. Z-index values are expressed as anomalies with respect to the 1961–1990 reference period (horizontal line). Points mark the driest/wettest years while grey areas identify the driest/wettest 10-year and 30-year periods respectively.





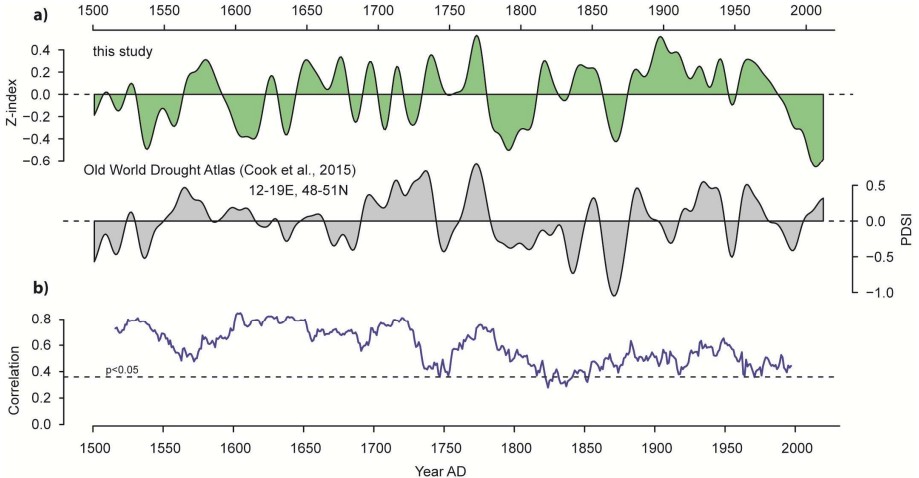

**Figure 6. (a) Long-term variability of AMJJ Z-index for the CR compared with scPDSI (Cook et al., 2015, areal mean of the 48–51°N; 12–19°E region) in the 1501–2012 period; original series smoothed by 30-year Gaussian filter; (b) 30-year running correlations between the Z-index and PDSI series; correlations above the dashed horizontal line are statistically significant**




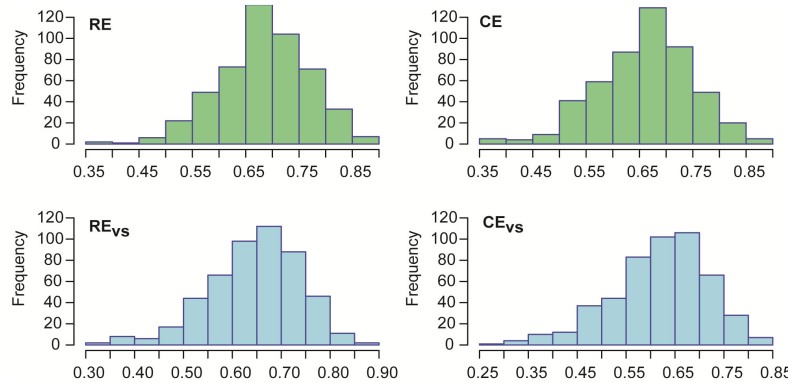

**Figure 7. Frequency distribution of RE, CE, $RE_{vs}$ and $CE_{vs}$ verification statistics calculated from 500 different realizations of 25 calibration and 25 verification years randomly selected from the 1805–1854 period**



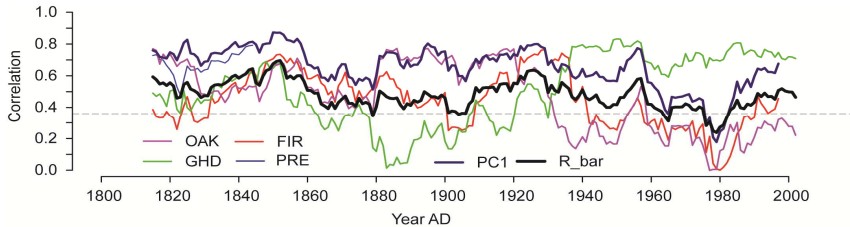

**Figure 8. 21-year running correlations between instrumental Z-index (target data), individual proxy series (OAK, FIR, GHD, PRE) and their PC1 in the 1805–2012 period; R-bar denotes the mean correlation and the horizontal dashed line is level of**
5  **significant positive correlation (p <0.05)**



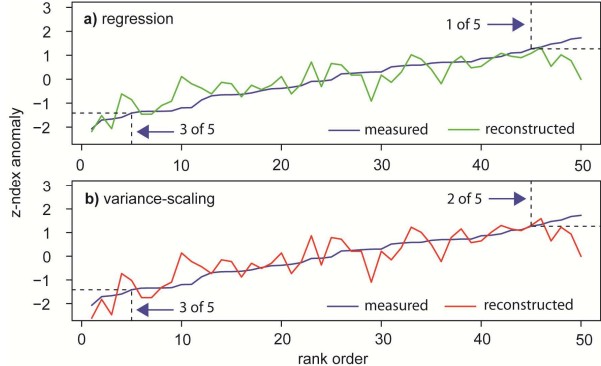

**Figure 9. Comparison of measured Z-index (continuous blue line) and reconstructed Z-index using (a) regression (green line) and (b) variance-scaling (red line) presented in rank order. Dashed boxes indicate the lower (dry) and upper (wet) 10% of instrumental Z-index values and numbers of these extremes reproduced in reconstructions.**