# Peer review of "A 500-year multi-proxy drought reconstruction for the Czech Lands"

_Climate of the Past, 2018_

## Referee Comment (RC1) · McCarroll (Referee) · 29 Jan 2019

This paper combines some previously published, and very powerful hydroclimate proxies to produce a multi-proxy reconstruction of April to July drought variability (Palmer moisture anomaly index). However, the variable length of the proxy series, and the virtual collapse of signal for some of the longer ones in the 20th Century, means that the calibration and verification rests on a very short period of common overlap with instrumental measurements. Given the way that the data have been treated, and the very short and early calibration period, I am not convinced that the conclusions of this paper are robust. The individual series have already been used to produce excellent palaeoclimate reconstructions and I see limited value in this compilation. I list my main concerns.

[Figure]

1. The short and early calibration/verification period is clearly acknowledged by the authors and they have performed a wide range of analyses to test it. However, I strongly suspect that their methods, when applied to such a short period, are resulting in over-fitting of a model to target data, and the available instrumental data are not sufficient to really test that.

Short calibration periods are a serious problem, and for some proxies, particularly historical data, they are unavoidable. This team has led the way in dealing with those problems in their earlier papers. However, when proxies are combined using multivariate methods, such as those used in this paper, the problems are greatly multiplied. When we only have a single proxy the relationship between the proxy and the target is defined over the calibration period and then we re-scale the proxy using (typically) either regression or variance scaling. The strength of the correlation gives us an indication of the uncertainty in our scaling parameter. However, with several proxies we are not just dealing with uncertainty in scaling, but in uncertainty in the WEIGHT of each proxy in defining that scaling. In this paper the weights are defined over a very short period using a combination of step-wise regression and PCA analysis. With such a short period of overlap it is not possible to test (robustly) whether the weights are stable over time. Using PCA of several proxies it is almost inevitable that the model will produce a composite (a model) that correlates well with the target, the question is whether an independent period would produce a model with similar weightings.

The evidence presented in the paper strongly suggests that if the data could be extended into the more recent period a model would NOT have the same weightings, because the quality of some of the proxies deteriorates (for good reason probably). Before the calibration period there are certainly intervals where the various proxies do not correlate with each other very well and where the relative strength of between-proxy correlations change over time (Fig. 2). This gives me little confidence that the weightings in the model are likely to be consistent over time, which likely means that the model presented is over-fitted to the data. That is why it explains so much of the

variance.

2. The authors try to test the validity of the short calibration period using standard calibration and verification tests, but this is not a good test in this case. First it is not clear to me whether the whole procedure of building the model and defining the weights is conduced independently over the two 25yr periods. It is not useful to just compare the PC values defined over the whole period. If the models have been computed independently then give the weightings.

Even if the models have been computed completely independently, comparing two contiguous 25yr periods is a very weak test, and especially so in this case because the mean value of the Z-index appears to be almost the same across the two periods. If this is the case then the RE and CE tests are not much use. The CE test in particular is used specifically to test the ability of a model to capture a change in level, and if there is no change in level it is redundant. Overall they are used to test for temporal coherence and 25 years is just too short to test that.

The multiple re-sampling of 25 years out of 50, without replacement (Fig. 7) is pointless. If the model is over-fitted (I mean the weights of the proxies) and the overall correlation over 50 years is therefore very strong, then even taking the 25 years with the highest squared error (that is the worst possible selection) will still result in a high correlation- otherwise the overall correlation could not be that high.

The critical point here is that the statement (p. 6 line 32) that 'this demonstrates that the hydroclimate signal included in the PC1 scores transformed from the original four proxy series is strong and stable over time' is not robust. Similarly the statement that the 95% error bars do not have to be adjusted because the R-bar before 1800 (p. 7 l5) is not robust because it assumes that the weightings attributed to the 4 proxies in the PC are perfect. It is the uncertainty in those weights that will contribute most to the uncertainty of the reconstruction.

3. The authors compare the regression and variance scaled approaches using various

metrics. Given the very high correlation between PC1 and the target that is not surprising. As r approaches one then the regression and variance scaled reconstructions become effectively the same thing. The critical test in this case is the capture of extremes. Hydroclimate reconstructions that do not capture hydroclimate extremes are of limited value. In this case, despite the very high r-value, only one of the 5 wettest years was captured by the regression model. There seems to be an asymmetry in the ability of the models to capture extremes- they are better at dry years than wet ones. That is critically important for reconstructing the past, given the dire consequences of runs of wet summers for agrarian societies.

4. The conclusions of this paper relate to the most severe periods of hydroclimate in the past 500 years. That is the bit that will be of most interest to the wider scientific community. However, what are the consequences of the de-trending of the data sets for this analysis?

I struggle to detect exactly how much de-trending has been applied to the individual data sets, and in figure 2 it says 'smoothed by a low-pass filter'. In the abstract it says 'the new Z-index reconstruction does not capture any long-term trends beyond the multi-decadal domain'. I take this to mean that the low-frequency information has been lost. How then is it possible to come to the main conclusion of the paper, which is that 'Central Europe has recently experienced the most severe decadal-scale. . . drought of the last 500-years'.

I presume that the term 'decadal scale' is supposed to convey something about the loss of low-frequency, but that is going to be lost on most readers and I am not sure exactly what it means. What we really want to know is whether the recent drought is more extreme than any drought of similar magnitude in the last 500 years. Just as we want to know whether 21st century temperatures are similarly extreme. If the proxy data have been de-trended then that information has been removed. The use of phrases like 'decadal scale' is not helpful, it is obscuring the problem, not helping to illuminate it.

In conclusion I am afraid I am not in favour of publishing this paper. I strongly suspect that it is based on over-fitting a multivariate model to a calibration that is too short. The result is an explained variance that is unrealistically high and error bars that are far too optimistic.

Of most concern is the strong conclusions that are drawn in relation to the extremity of recent decades. This kind of conclusion is critically important for the study of climate change, and as a community we have to be confident that when we say 'the recent decades are really unusual' that we are crystal-clear about the evidence and the uncertainty. In this case, the use of multivariate methods applied over such a short and early calibration period, and the influence of de-trending, make me very nervous about coming to such a firm conclusion. Adding phrases like 'decadal scale' is really not helpful to anyone.

The proxy series on which this paper is based have already been published, and the documentary and grape-harvest data in particular have provided beautiful climate reconstructions. Those reconstructions are not perfect, but the methods used to produce them are transparent and the strengths and weaknesses are reasonably clear to the rest of the community. I strongly feel that by combing these 4 proxy series nothing has really been added. The multi-proxy reconstruction appears very strong, but it is probably just over-fitted to the short calibration period, and the multivariate analysis obscures the relative contributions of the different data sets, so that it is much more difficult to assess strengths and weaknesses of the final reconstruction.

I am sorry to be negative about such a well prepared and well written manuscript by such a strong team.

---

## Referee Comment (RC2) · Anonymous Referee #2 · 6 Feb 2019

Dobrovolny et al. presents a new multiproxy hydroclimate reconstruction for the Czech Republic spanning back to 1501 AD, based on four different proxy types: tree-ring width chronologies from oak and silver fir, grape harvest dates and historical documentary evidence. While most of these records are subsets of published data that have previously been used in single proxy reconstructions to infer past hydroclimate/rainfall variability in the region, no attempts to synthesize moisture-sensitive proxies with different climate retention characteristics have, to my knowledge, yet been made for this particular region. This type of work has therefore the potential to be compelling and of scientific value for the paleoclimate community, as it has the possibility to provide a more accurate reconstruction at a range of time scales and also help to quantify the uncertainty in the individual proxies in a rigorous manner. Given this potential, I found

however some of the methods selected in the current study not entirely justified. I also found the manuscript to be a bit untidy when it comes to the details. There are thus a few very important points that would be strengthened through revision.

It is not obvious in the manuscript what additional information that are gained by combining the different kinds of proxies into a single hydroclimate reconstruction compared to the previous published reconstruction efforts using the individual proxy records separately? For example, the abstract concludes "The new multi-proxy drought reconstruction demonstrated progress beyond previous single-proxy attempts at establishing the strength of hydroclimate signal." The authors should be more explicit, what is the novel contribution of this new reconstruction and in what way does it move the field forward? The explained variance of 70% is indeed impressive, but as far as I can see not superior to the amount of variance explained when using the GHD record separately to reconstruct regional SPEI (P3/L31). Also, the target season seems to be shortened when combining multiple proxy records. Another example, page 2/L21 reads "To overcome some of the problems of the single-proxy approach, to identify possible seasonal biases and finally to provide more rigorous estimates of past and present hydroclimate variability, this contribution attempts to synthesize available precipitation-sensitive proxies from the territory of the CR and to utilize them in the first multi-proxy reconstruction of droughts in the area." These objectives does not match the rest of the paper; I cannot see that the authors have identified and provided a solution to the seasonal mismatch among the proxies. Neither is there any in-depth test or even discussion provided on the advantage of the current multi-proxy reconstruction compared to the previous single-proxy attempts. Perhaps is there an oversight from my part, but the only reconstruction uncertainty estimates that I can find in this work are based on the regression error (figure 5). I suggest that the authors either rephrase the introduction/objective section to match the methods and the results, or explicitly identify the problems of the single-proxy approach and state what test they have designed in this study to overcome these issues.

My other concern relates to the very short calibration/verification period used to build the reconstruction model. Reviewer no. 1 has already raised some critical points concerning this issue, for example the possibility for overfitting the model which has the effect of underestimating the prediction error. Another potential problem is the fact that early meteorological observations are utilized for the calibration/verification exercise. Because of the scarcity of observations, the early instrumental data cannot usually be satisfyingly checked for their quality and homogeneity (even for a leading climate parameter such as temperature, and for a region such as central Europe). Before applying the early instrumental data in proxy calibration it should be considered to what extent they reflect real climate (homogeneity) and to what extent the observations are representative for the proxy sites (spatial representation). It is not clear how these issues have been addressed in the current work. Perhaps an oversight from my side, but I cannot find any information in the manuscript on which and how many stations the authors have used in the current work to compute the Z-index, which selection criteria were applied? Is the calibration data some kind of a regional mean for the whole domain? If so, how many stations are included and where are they located (I would suggest to add their location to fig 1)? Is the number of stations changing through time? These issues may potentially have an important implication the calibration/verification statistics and should be at least be mentioned in the text.

Some of the details in the data and method sections are at times rather swiftly handled and details are missing that would help the readership to understand the rationale for why certain approaches were taken. Specifically: P2: How were the different tree-ring data standardized? From eyeballing fig 2 I get the impression that all four proxy time series contain different amounts of low and high frequency variability, autocorrelation and sometimes the variance is conspicuously instable (the GHD chronology). How is the final reconstruction affected by these issues? This is an important problem that must be considered when applying a multi-proxy approach. Please provide at least a discussion around this issue. L1/P22: "While reflecting a considerable amount of inter-annual spring-summer drought variability, the new Z-index reconstruction does not capture any

long-term trends beyond the multi-decadal domain." Reading this paragraph the less well-informed readership might get the impression that the region has not experienced any long-term changes in hydroclimate beyond the multi-decadal domain, whereas the absence of any long-term trend in the reconstruction in fact is more likely related to the limited ability of the proxy data to encode climate variability at the centennial and longer time scales. Please rephrase to make this clearer.

I have noted a few instances where the text requires revising or minor revision to clarify particular points, see my comments below. I hope my comments will help the authors to improve the manuscript.

P2/L4: change to " will become" P2/L5: ". . . partly corrected with hydroclimate proxy reconstructions" I am not sure that "corrected" is the right word to use in this context P4/L21: clarify over which period the PCA was performed, is it performed separately over the calibration and verification periods? Also, I cannot find in the ms how the different chronologies are loaded on the principal component, which chronologies contribute the most to the reconstruction and is the contribution stable over the calibration and verification intervals? P5/L1: "The mean of the target data was added to the PC scores to ensure the same mean for both proxy and target". Over which period were the statistics computed? P5/L26: "tendencies" – change to variability or trend (suggestion) P6/L18: please check that the R2 value reported is consistent throughout the ms, for example P1/L19 – 70%, P6/L10 – 67.8%, P6/L18, 69.7%, table 1. P7/L21: "The driest year (1616) had already been identified in numerous studies analyzing documentary evidence (Brázdil et al., 2013b; Dobrovolný et al., 2015) and tree-ring chronologies (Dobrovolný et al., 2018) covering CR territory." Yes, but are these data independent from the one used in the current study? At least the tree-ring chronologies in Dobrovolný et al., 2018 seems to overlap with the current study. P8/L19: It is unclear from the text whether the Old World Drought Atlas and the current reconstruction share some of the predictors. Table 1: full calibration – do you mean 1805–1854 ? Figure 1: I would suggest adding the location of the meteorological stations to the map. Figure

5: is this the regression based reconstruction or the variance scaled?